# Metabolomic Analysis of the Effect of Freezing on Leaves of *Malus sieversii (Ledeb.) M.Roem.* Histoculture Seedlings

**DOI:** 10.3390/ijms25010310

**Published:** 2023-12-25

**Authors:** Yongfeng Su, Lijun Liu, Hongxi Ma, Yinyan Yuan, Deen Zhang, Xiaoyan Lu

**Affiliations:** Xinjiang Production and Construction Corps Key Laboratory of Special Fruits and Vegetables Cultivation Physiology and Germplasm Resources Utilization, Agricultural College of Shihezi University, Shihezi 832003, China; 18193903616@163.com (Y.S.); m13094043010_1@163.com (L.L.); 17590396025@163.com (H.M.); yuanyinyan2022@163.com (Y.Y.); 18899129008@163.com (D.Z.)

**Keywords:** *Malus sieversii (Ledeb.) M.Roem.* histoculture seedlings, freezing stress, metabolomics

## Abstract

*Malus sieversii (Ledeb.) M.Roem.* is the ancestor of cultivated apples, and is an excellent germplasm resource with high resistance to cold. Artificial refrigerators were used to simulate the low temperature of −3 °C to treat *Malus sieversii (Ledeb.) M.Roem.* histoculture seedlings. Observations were performed to find the effects of freezing stress on the status of open or closed stomata, photosystems, and detection of metabolomic products in leaves of *Malus sieversii (Ledeb.) M.Roem.* histoculture seedlings. The percentage of closed stomata in the *Malus sieversii (Ledeb.) M.Roem.* histoculture seedlings increased, the maximum fluorescence (Fm’) excited by a strong light (saturating pulse) was weakened relative to the real-time fluorescence in its vicinity, and the quantum yield of unregulated energy dissipation was increased in PSII under freezing stress. The metabolites in the leaves of the *Malus sieversii (Ledeb. M.Roem.)* histoculture seedlings were analyzed by ultra-performance liquid chromatography–tandem mass spectrometry using CK, T12h, T36 h, and HF24h. Results demonstrated that cold stress in the *Malus sieversii (Ledeb.) M.Roem.* histoculture seedlings led to wilting, leaf stomatal closure, and photosystem damage. There were 1020 metabolites identified as lipids (10.2%), nucleotides and their derivatives (5.2%), phenolic acids (19.12%), flavonoids (24.51%), amino acids and their derivatives (7.75%), alkaloids (5.39%), terpenoids (8.24%), lignans (3.04%), organic acids (5.88%), and tannins (0.88%). There were 110 differential metabolites at CKvsT12h, 113 differential metabolites at CKvsT36h, 87 differential metabolites at T12hvsT36h, 128 differential metabolites at CKvsHF24h, 121 differential metabolites at T12hvsHF24h, and 152 differential metabolites at T36hvsHF24h. The differential metabolites in the leaves of the *Malus sieversii (Ledeb.) M.Roem.* seedlings grown under low-temperature stress mainly involved glycolysis, amino acid metabolism, lipid metabolism, pyrimidine metabolism, purine metabolism, and secondary metabolite metabolism. The *Malus sieversii (Ledeb.) M.Roem.* seedlings responded to the freezing stress by coordinating with each other through these metabolic pathways. The metabolic network of the leaves of the *Malus sieversii (Ledeb.) M.Roem.* histoculture seedlings under low temperature stress was also proposed based on the above pathways to deepen understanding of the response of metabolites of *Malus sieversii (Ledeb.) M.Roem.* to low-temperature stress and to lay a theoretical foundation for the development and utilization of *Malus sieversii (Ledeb.) M.Roem.* cultivation resources.

## 1. Introduction

*Malus sieversii (Ledeb.) M.Roem.* is the main ancestor of many domesticated apple varieties (M. domestica), mainly in Central Asia and Xinjiang, China [1,2]. *Malus sieversii (Ledeb.) M.Roem.* is a germplasm resource with excellent cold resistance traits and is usually used as a popular rootstock for breeding cold-resistant domesticated apples in China [3,4].

Cold stress is one of the chief abiotic stresses that can adversely affect the physiology and development of the plant and crop yield [5]. When temperatures deviate from the range of optimal survival values, plants can experience a severe degree of physiological, cellular, metabolic, and molecular dysfunction that can lead to growth cessation and ultimately to death [6]. The potential symptoms of chilling injury are surface lesions, a water-soaked appearance of the tissues, water loss/desiccation, internal discoloration, and tissue breakdown [7]. In response to adverse environments, plants typically alter gene transcription, antifreeze protein expression, and increased levels of antifreeze-related metabolites [8,9].

The analysis of plant metabolism under stress can be greatly aided by metabolomics, which can also be used to determine the products of gene transcription and protein modification and more accurately depict the regulatory mechanisms of cells in signal transduction and energy transmission [10]. Although the statistics are incomplete, the total number of metabolites in plants is approximately 200,000 [11,12]. Changes in the variety and number of metabolites can reflect the adaptability of plants to the environment [13,14]. The freezing-tolerant and -sensitive Arabidopsis ecotypes Wassilewaskija-2 (Ws-2) and Cape Verde Islands-1 (Cvi-1) were compared for alterations brought on by stress using GC-TOF-MS analysis, respectively. With the discovery of several metabolites that were first described as accumulating during cold acclimation in Arabidopsis, the study proved the effectiveness of GC-MS-based metabolite profiling [15]. In the study of asparagus under low temperature stress, the results showed that the differential metabolites were mainly sugars, amino acids, organic acids, lipids, nucleotides, phenolic acids and vitamins [16]. In higher plants, glucose, sucrose, fructose, raffinose, and stachyose are known to have freeze tolerance [17]. Studies have shown that soluble sugar levels in different plants increase significantly under low-temperature stress, such as in citrus (Citrus reticulata) [18], red spruce [19], and Pinus halepensis [20]. Many cold-resistant proteins and protective substances (e.g., soluble sugar and proline) are synthesized in plant cells, which regulate osmotic potential and maintain membrane integrity [17,21]. Under low-temperature stress, arginine, proline, glutamine, asparagine, and other amino acids in pomegranate are relatively high, especially alanine, tryptophan, histidine, and other amino acids. The accumulation of these amino acids has important significance for improving the cold resistance of pomegranate [22].

Metabolomics are an effective method to explain plant growth and adaptation mechanisms in harsh environments, and have been widely used to investigate plant response to abiotic stresses [23,24]. Studies have shown that few studies have reported metabolic changes in *Malus sieversii (Ledeb.) M.Roem.* in response to low-temperature stress. Using artificially modified refrigerators to simulate low-temperature environments to stress *Malus sieversii (Ledeb.) M.Roem.* seedlings, changes in metabolites in the leaves of the *Malus sieversii (Ledeb.) M.Roem.* seedlings were comprehensively analyzed by ultra-performance liquid chromatography–tandem mass spectrometry (UPLC–MS/MS), so as to understand the response of the metabolites of *Malus sieversii (Ledeb.) M.Roem.* to low-temperature stress and lay the foundation for the utilization of *Malus sieversii (Ledeb.) M.Roem.* germplasm resources.

## 2. Results

### 2.1. Effect of Freezing on Morphology and Leaf Stomata of Malus sieversii (Ledeb.) M.Roem. Histoculture Seedlings

Freezing stress prolongs the duration of wilting in *Malus sieversii (Ledeb.) M.Roem.* histoculture seedlings, as related studies show in the previous research [25]. The stomata of the *Malus sieversii (Ledeb.) M.Roem.* histoculture seedling leaves were observed morphologically under freezing stress (Figure 1a). It was discovered that the *Malus sieversii (Ledeb.) M.Roem.* seedling leaves had 88% of their stomata open at CK; at T12h, they had 23% of their stomata open; at T36h, they had 31% of their stomata open; and, at HF24h, they had 84% of their stomata open (Figure 1b).

### 2.2. Effect of Freezing on Chlorophyll Fluorescence of Malus sieversii (Ledeb.) M.Roem. Histoculture Seedlings

Chlorophyll fluorescence induction curves were observed in the *Malus sieversii (Ledeb.) M.Roem.* histoculture seedlings. When the real-time fluorescence of the *Malus sieversii (Ledeb.) M.Roem.* histoculture seedlings stabilized, the jump in maximum fluorescence (Fm’) excited by a strong light (saturating pulse) weakened with the prolongation of the freezing stress time (CK, T12h, T36h); there was a significant recovery at HF24h (Figure 2a), with the Y(NO) indicating the quantum yield of the non-regulatory energy dissipation at the PSII. The color of fluorescence imaging changed from yellow to blue with increasing freeze stress. at HF24h, the color changed from blue to yellow (Figure 2b). The values of Y(NO) were high, indicating that the photochemical energy conversion and protective regulatory mechanisms were impaired. The Y(NO) values for the *Malus sieversii (Ledeb.) M.Roem.* histoculture seedlings under freezing stress increased with the prolongation of freezing time (Figure 2c), which appealed to the fact that the *Malus sieversii (Ledeb.) M.Roem.* histoculture seedlings suffered deeper damage with the prolongation of freezing stress.

### 2.3. Metabolomics Data Analysis

#### 2.3.1. Metabolite Classification of Leaf Metabolites in *Malus sieversii (Ledeb.) M.Roem.* Histoculture Seedlings under Freezing Stress

Qualitative analysis of metabolites in the leaves of the *Malus sieversii (Ledeb.) M.Roem.* histoculture seedlings under freezing stress at different times was performed, and a total of 1020 metabolites were identified in the samples. Based on the Kyoto Encyclopedia of Genes and Genomes (KEGG) compound database, the MetWare database (MWDB), and multiple reaction monitoring (MRM) techniques, the 1020 metabolites identified were classified into lipids (10.2%), nucleotides and their derivatives (5.2%), phenolic acids (19.12%), flavonoids (24.51%), amino acids and their derivatives (7.75%), alkaloids (5.39%), terpenoids (8.24%), lignans (3.04%), organic acids (5.88%), and tannins (0.88%) with flavonoid metabolites accounting for the highest percentage (Figure 3).

#### 2.3.2. Qualitative Analysis of Leaf Metabolites in *Malus sieversii (Ledeb.) M.Roem.* Histoculture Seedlings under Freezing Stress

Principal Component Analysis (PCA) is a multidimensional statistical data analysis method of unsupervised pattern recognition. Through principal component analysis of the samples, we can preliminarily understand the overall metabolic differences among samples and the degree of variability between samples within groups. According to the analyses, the variations between the CK, T12h, T36h, and HF24h groups were significantly larger than the variations within the groups (Figure 4).

Although PCA can effectively extract primary information, it is insensitive to variables with small correlations, which can be addressed by partial least squares-discriminant analysis (PLS-DA). Compared with PCA, PLS-DA can maximize the differentiation of groups and facilitate the search for differentiated metabolites. OPLS-DA modeling first excludes orthogonal variables that are not related to metabolite categorical variables, and then analyzes the association differences between and within groups. Using the OPLS-DA model, the metabolite group data were analyzed and the scores of each group were plotted (Appendix A) to further show the differences between the groups. The prediction parameters of the OPLS-DA evaluation model were R2X, R2Y, and Q2. R2X and R2Y represent the explanation rate of the model for the X and Y matrices, respectively, and Q2 indicates the predictive ability of the model. The closer these three indicators are to 1, the more stable and reliable the model is. Q2 > 0.5 can be regarded as an effective model, and Q2 > 0.9 is an excellent model. In each OPLS-DA evaluation model, Q2 and R2Y were greater than 0.9, and R2X was greater than 0.5, which shows that the model is reliable. From the results, there are obvious differences between the four groups of data (CK, T12h, T36h, and HF24h) (Appendix A). The differentiated metabolites could be screened according to the VIP value analysis (Appendix A).

#### 2.3.3. Metabolite Differential Analysis

When screening for differential metabolites by combining variable importance in projection (VIP) and ploidy change for all 6 pairwise comparisons, 1020 metabolites were detected in the leaves of the *Malus sieversii (Ledeb.) M.Roem.* histoculture seedlings. There were 110 differential metabolites at CKvsT12h, 113 differential metabolites at CKvsT36h, 87 differential metabolites at T12hvsT36h, 128 differential metabolites at CKvsHF24h, 121 metabolic differentials at T12hvsHF24h, and 152 differential metabolites at T36hvsHF24h (Figure 5a–f). The number of different metabolites in different species is shown (Table 1).

After searching for all differential metabolites in pairwise comparisons, and then searching for common differential metabolites of CKvsT12h, CKvsT36h, and T12hvsT36h, 32 common differential metabolites were found and a total of 174 differential metabolites were identified in the three comparisons of CKvsT12h, CKvsT36h, and T12hvsT36h. A total of 48 common differential metabolites were identified for CKvsHF24h, T12hvsHF24h, and T36hvsHF24h, and a total of 234 differential metabolites were identified in the three comparisons of CKvsT12h, CKvsT36h, and T12hvsT36h. In the 6 comparisons of CKvsT12h, CKvsT36h, T12hvsT36h, CKvsHF24h, T12hvsHF24h, and T36hvsHF24h, with 408 differential metabolites (Figure 5g), 280 differential metabolites were obtained after removing duplicates of the 408 differential metabolites (the same differential metabolite existed with different comparisons).

A K-Means analysis was performed to determine trends in differential metabolites associated with low-temperature stress at different periods. The results show that these trends can be divided into five categories, aggregating 25, 32, 40, 89, and 94 metabolites in sub classes 1 to 5, respectively. In sub class 1, the change in metabolites decreased from CK to the middle of T12h and fluctuated up and down from T12h to HF24h. In subclass 2, the change in metabolites increased from CK to HF24h, and in subclass 3 there were 40 metabolites, which showed an increasing trend from CK to T12H, a decreasing trend from T12H to T36H, and a decreasing trend between T36h and HF24h for some metabolites and an increasing trend for the others. In subclass 4, there were 89 metabolites, and the trend from CK to T36H was neither decreasing nor increasing, and the trend between T36h and HF24h was increasing. In subclass 5, with 94 metabolites, the trend from CK to T36H for these 94 metabolites was increasing, and the trend between T36h and HF24h was decreasing (Figure 5i).

#### 2.3.4. Integrated Metabolic Network Analysis of *Malus sieversii (Ledeb.) M.Roem.* Histocultures under Freezing Stress

According to the KEGG pathway database, differential metabolites were enriched in different pathways, and differential metabolites for different pairwise comparisons involved carbon metabolic pathways, secondary metabolite biosynthesis, amino acid biosynthesis, linolenic acid and α-linolenic acid metabolism, purine metabolism, pyrimidine metabolism, and vitamin 6B metabolic pathways. To gain a comprehensive understanding of metabolite changes in *Malus sieversii (Ledeb.) M.Roem.* group embryos under simulated low-temperature stress, we proposed a metabolic pathway containing each of these modules to elucidate metabolite changes in *Malus sieversii (Ledeb.) M.Roem.* histoculture seedlings corresponding to freezing stress.

Fourteen metabolites were identified as involved in glycolysis, tricarboxylic acid cycle, and pentose phosphate cycle, including sucrose, glucose-1-phosphate, D-glucose, D-glucose-6-phosphate, D-fructose-6-phosphate, D-Trehalose, gluconic acid, citric acid, isocitric acid, succinic acid, fumaric acid, L-malic acid, L-aspartic acid, α-ketoglutaric acid, and L-glutamic acid. D-Trehalose was significantly upregulated at T12hvsHF24h; gluconic acid was significantly upregulated at CKvsT12h, CKvsT36h, and CKvsHF24h; D-fructose-6-phosphate* and glucose-1-phosphate* were significantly upregulated; α-ketoglutaric acid and L-glutamic acid were significantly upregulated at T36hvsHF24h; L-aspartic acid was significantly upregulated at CKvsT36h; D-glucose was significantly downregulated at CKvsT36h; α-ketoglutaric acid was significantly downregulated at T12hvsT36h; and L-glutamic acid was significantly downregulated at CKvsT36h (Figure 6a).

Metabolites associated with metabolic pathways for vitamins are pyridoxal, pyridoxine, 4-pyridoxic acid, and ribulose-5-phosphate. Pyridoxal was significantly downregulated at CKvsT36h, and ribulose-5-phosphate was significantly upregulated at T12hvsT36h. The metabolites associated with pyrimidine metabolism were cytidine, cytosine, uracil, uridine, and 5′-diphosphate-D-glucose. Uracil was significantly upregulated at CKvsT36h, and uridine, cytidine and cytosine were significantly upregulated at CKvsHF24h, T12hvsHF24h, and T36hvsHF24h. The metabolites associated with purine metabolism were Inosine-5′-monophosphate, Guanosine-5′-monophosphate, Guanosine-3′,5′-cyclic-monophosphate, Guanosine, Guanine, Xanthosine, Inosine, Adenosine*, Adenine, Adenosine-5′-monophosphate,2′-Deoxyadenosine-5′-monophosphate,2′-Deoxyadenosine,2′-Deoxyinosine, and Cyclic-3′,5′-Adenylic acid. Inosine, Adenosine, and Cyclic-3′,5′-Adenylic acid were significantly upregulated at CKvsHF24h; Guanosine-(3′,5)′-cyclic-monophosphate and Guanine were significantly upregulated at T12hvsHF24h and T36hvsHF24h; Guanosine was significantly upregulated at CKvsHF24h, T12hvsHF24h, and T36hvsHF24h; Xanthosine was significantly upregulated at T12hvsT36h, CKvsHF24h, and T12hvsHF24h; Guanosine-3′,5′-cyclic-monophosphate was significantly downregulated at T12hvsT36h (Figure 6b).

The main metabolites involved in amino acid synthesis are Valine, 2-Isopropylmalic Acid, 3-Isopropylmalic Acid, L-Leucine, 3-Methylmalic acid, L-Isoleucine, Citric Acid, Isocitric Acid, α-Ketoglutaric acid, L-Glutamine, L-Glutamicacid, N-Acetyl-L-ornithine, L-Ornithine, L-Citrulline, Argininosuccinic acid, L-Arginine, L-Proline, L-Aspartic Acid, L-Asparagine, L-Homoserine, L-Threonine, L-Serine, L-Homocysteine, L-Methionine, S-(5′-Adenosyl)-L-methionine, S-(5′-Adenosy)-L-homocysteine,3-Dehydroshikimic acid, Anthranilic Acid, L-Tryptophan, L-Tyrosine, Phenylpyruic acid, and L-Phenylalanine. Valine, L-Isoleucine, L-Glutamine, and L-Leucine were significantly downregulated at T36hvsHF24h; L-Serine, S-(5′-Adenosy)-L-homocysteine, and L-Glutamic acid were significantly downregulated at CKvsT36h; S-(5′-Adenosyl)-L-methionine was significantly downregulated at T12hvsT36h; L-Ornithine was significantly downregulated at CKvsHF24h, T12hvsHF24h, and T36hvsHF24h; L-Arginine and L-Threonine were significantly upregulated at CKvsT36h; L-Glutamic acid was significantly upregulated at T36hvsHF24h; 2-Isopropylmalic Acid was significantly upregulated at CKvsT12h, CKvsT36h, T12hvsT36h, and T12hvsHF24h; 3-Isopropylmalic Acid* was significantly upregulated at CKvsT12h, T12hvsT36h, and T12hvsHF24h; S-(5′-Adenosy)-L-methionine was significantly upregulated at CKvsT12h, CKvsT36h, CKvsHF24h, and T36hvsHF24h; S-(5′-Adenosy)-L-homocysteine was significantly upregulated at CKvsHF24h, T12hvsHF24h, and T36hvsHF24h (Figure 6c).

The metabolites associated with linolenic acid metabolism were Linoleic acid, γ-Linolenic Acid, Crepenynic acid, (9,10,13)-Trihydroxy-11-Octadecenoic Acid, 9S-Hydroxy-10E, and 13S-Hydroperoxy-9Z. Linoleic acid and 9S-Hydroxy-10E were significantly upregulated at CKvsT12h; (9,10,13)-Trihydroxy-11-Octadecenoic Acid was significantly upregulated at CKvsT12h, CKvsT36h, T12hvsT36h, and CKvsHF24h; 13S-Hydroperoxy-9Z was significantly upregulated at CKvsT12h and CKvsT36h; Crepenynic acid was significantly downregulated at T12hvsT36h; Linoleic acid was significantly downregulated at CKvsT12h, T12hvsT36h, and T12hvsHF24h; and 9S-Hydroxy-10E was significantly downregulated at T12hvsT36h and T12hvsHF24h. Metabolites related to α-linolenic acid metabolism include α-Linolenic Acid, 17-Hydroxylinolenic acid, 13(s)-hydroperoxy-(9z,11e,15z)-octadecatrienoic acid, 2R-hydroxy-(9Z,12Z,15Z)-octadecatrienoic acid, 13S-Hydroxy-(9Z,11E,15Z)-octadecatrienoic acid, 9-Hydroxy-(12-oxo-10E,15Z)-octadecadienoic acid, and Jasmonic acid. 17-Hydroxylinolenic acid, 2R-hydroxy-(9Z,12Z,15Z)-octadecatrienoic acid, and 9-Hydroxy-(12-oxo-10E,15Z)-octadecadienoic acid were significantly upregulated at CKvsT12h; Jasmonic acid was significantly upregulated at T36hvsHF24h; 17-Hydroxylinolenic acid was significantly downregulated at T12hvsT36h and T12hvsHF24h; and 13S-Hydroxy-(9Z,11E,15Z)-octadecatrienoic acid was significantly downregulated at T12hvsT36h (Figure 6d).

Metabolites involved in the phenylpropanoid biosynthetic pathway include L-Phenylalanine, L-Tyrosine, Ferulic acid, Sinapic acid, 1-O-Sinapoyl-β-D-glucose, Sinapoyl malate, Cinnamic acid, Caffeic acid, Trans-5-O-(p-Coumaroyl)shikimate,5-O-Caffeoylshikimic acid, 5-O-p-Coumaroylquinic acid*, Chlorogenic acid, Coniferyl alcohol, Caffeic aldehyde, and Coniferin. Cinnamic acid, Caffeic acid, Ferulic acid, and Sinapic acid were significantly upregulated at CKvsHF24h, T12hvsHF24h, and T36hvsHF24h; 5-O-Caffeoylshikimic acid was significantly upregulated at CKvsT12h, CKvsHF24h, T12hvsHF24h, and T36hvsHF24h; 1-O-Sinapoyl-β-D-glucose was significantly upregulated at CKvsHF24h and T36hvsHF24h; 5-O-p-Coumaroylquinic acid* was significantly upregulated at CKvsT36h; Coniferyl alcohol was significantly upregulated at T12hvsT36h; and Sinapic acid was significantly downregulated at T12hvsT36h (Figure 6e).

Metabolites involved in the flavonoid biosynthetic pathway include Luteolin, Cosmosiin, Cynaroside, Luteolin-7-O-glucuronide, Lonicerin, Afzelin, Trifolin, Astragalin, Quercetin, Nicotiflorin, Quercetin-3-O-(6″-O-malonyl)glucoside, Quercetin-3-O-sambubioside*, Rutin, Baimaside, Kaempferol, and Myricetin. Luteolin was significantly upregulated at CKvsT12h, T12hvsT36h, and T12hvsHF24h. Kaempferol and Rutin were significantly upregulated at CKvsT36h. Quercetin was significantly upregulated at T12hvsHF24h and T36hvsHF24h. Metabolites involved in the metabolic pathway of flavonoid and chalcone biosynthesis include Butin, Phloretin, Fustin, Phlorizin, Naringenin chalcone, Naringenin, Luteolin, Epicatechin, Prunin, Hesperetin-7-O-glucoside, Neohesperidin,(3,4,2’,4’,6’)-Pentahydroxychalcone, Taxifolin, Epiafzelechin, Epiafzelechin, and Catechin*. Taxifolin and Eriodictyol were significantly downregulated at CKvsT12h, CKvsHF24h, T12hvsHF24h, and T36hvsHF24h; Luteolin was significantly upregulated at CKvsT12h, CKvsHF24h, and T36hvsHF24h; Naringenin chalcone was significantly upregulated at T36hvsHF24h. Luteolin was significantly downregulated at T12hvsT36h (Figure 6e). 

## 3. Discussion

Cold stress is an environmental factor that has a significant impact on plant growth, productivity, and survival, and it limits the geographic distribution of various plant species. Cold temperatures (0–12 °C) inhibit plant growth and development, and prolonged exposure to freezing temperatures below 0 °C can damage cell membranes and lead to cell death [26,27,28].

Amino acid accumulation is a consistent response of plants to low temperatures and other abiotic stresses [29]. Amino acids are often considered compatible solutes that can accumulate at high concentrations for osmoregulation without disrupting cellular function. Various studies on cold-tolerant plants have shown that these compounds increase during acclimation. γ-aminobutyric acid (GABA) is a non-protein amino acid found in many organisms and is widely present in various parts of the plant body [30]. The GABA content in plants is relatively low under normal growth conditions, but the concentration increases dramatically under adverse conditions, such as biotic or abiotic stress, enhancing plant resistance by stabilizing the cell membrane structure, reducing reactive oxygen species damage, and regulating the synthesis of biomolecules [31]. Proline, putrescine, spermidine, and spermine contents were positively correlated with plant resistance to abiotic stresses [25,32,33,34,35,36,37,38,39,40,41,42,43,44,45,46,47,48,49,50,51,52,53,54,55,56,57,58,59]. Proline and the non-protein amino acid betaine are usually associated with low-temperature tolerance in herbaceous plants. In woody plants, proline and various other amino acids increase during domestication, and tryptophan shows continuous increases in conifers tolerant to extreme low temperatures such as white spruce, black spruce, ponderosa pine, and inverted spruce [60,61,62]. A large number of amino acids and their derivatives were accumulated in the leaves of the *Malus sieversii (Ledeb.) M.Roem.* histoculture seedlings under low-temperature stress, and they included L-aspartate-O-diglucoside, S-(5′-adenosyl)-L-methionine, 5-L-glutamyl-L-amino acid, L-leucine*, L-norleucine*, N-acetyl-L-leucine, L-isoleucine*, L-lysine butyrate, and γ-aminobutyric acid, which indicated that the types of amino acid accumulation under low-temperature stress were different in different plants.

Lipids accumulated under low-temperature stress included 4-hydroxysphingosine, 2-linoleoylglycerol-1-O-glucoside*, pomegranate, linoleic acid, transoleic acid*, carnosic acid*, n-eicosanoic acid 1-linoleoylglycerol*, ricinoleic acid, 17-hydroxylinolenic acid, 2-linoleoylglycerol*, 9S-hydroxy-10E,12Z-octadecadienoic acid*, 9-hydroxy-12-oxo- 10(E),15(Z)-octadecadienoic acid, 13-hydroxyoctadecyl-9,11-dienoic acid*, 2R-hydroxy-9Z,12Z,15Z-octadecatrienoic acid, 9,10,13-trihydroxy-11-octadecadienoic acid, 9,10,11-trihydroxy-12-octadecadienoic acid, 13-hydroperoxy-9,11-octadecadienoic acid, lysophosphatidylcholines, and lysophosphatidylethanolamines.

Plant leaves contain a variety of glycerolipid molecular species [33], and alteration of membrane lipid composition should be one of the ways in which plants respond to environmental stress. Mass spectrometry-based lipidomic platforms enable high-throughput analysis of the lipid species accumulated within the leaves of various plant species. Lipidomic analysis usually determines the total number of carbon atoms and acyl groups and double bonds within the glycerolipid molecule [34]. Arabidopsis leaves under normal growth conditions contain 18:3 (46–49 mol%), 18:2 (12–16 mol%), 16:0 (13–15 mol%), and 16:3 (14–16 mol%) [35].Lipid remodeling produces lipid intermediates such as lysophospholipids, which due to their water-soluble amphipathic and non-cylindrical structure, lysophospholipids and also affect lipid remodeling, membrane permeability, morphology, and stability [37]. Cold stress in Arabidopsis leaves under low-temperature stress contains 52:5-TAG (16:0/18:2/18:3) and 54:9-TAG (18:3/18:3/18:3) accumulation was higher [36]. Under low-temperature stress, LPE accumulated significantly in the walnut with strong low-temperature resistance compared with the walnut with weak resistance [56]. The significant accumulation of unsaturated fatty acids and lysophosphatidylcholines and lysophosphatidylethanolamines in the *Malus sieversii (Ledeb.) M.Roem.* histoculture seedlings under low-temperature stress indicates that the *Malus sieversii (Ledeb.) M.Roem.* histoculture seedlings undergo lipid remodeling to resist low-temperature stress.

The sugars accumulated under low-temperature stress mainly included gluconic acid, D-galacturonic acid, lycopene, D-pantotriose, sucrose-6-phosphate, and D-alginate. To combat abiotic stresses, plants have developed complex and well-coordinated molecular and metabolic networks in which sugar metabolism plays an important role [16], and the typical response of plants to abiotic stresses (e.g., salt, drought, and cold stresses) is the accumulation of sugars (e.g., glucose, fructose, sucrose, alginose, fructans, cottonseed sugars). Sugar accumulation may reflect osmoregulation, or in some cases provide osmoregulators, by maintaining cell swelling and play a role in ROS production and scavenging mechanisms [39]. The relationship between sugar and ROS can be indirect through the NADPH production metabolism, such as the oxidative pentose phosphate pathway, or direct with sugar acting as a true ROS scavenger [40]. Sucrose, cottonseed, and fructans have indeed shown to act as antioxidants in vitro by effectively quenching OH radicals [41,42], and sugars such as cottonseed, alginate, and fructans have also been suggested as protective agents for membrane integrity [43,44,45,46]. Under salt stress, the glucose and sucrose contents decreased slightly, while the fructose, xylose, galactose, and mannose contents increased [57].

Low-temperature stress induces a significant enrichment of organic acids in plant seedlings, and the level of organic acid content is an important indicator of plant growth and metabolic activity [25]. In a study on metabolites of *P. sylvestris* seedlings under low-temperature stress, TingTing Wang found that succinic acid, L-malic acid, citric acid, and fumaric acid were enriched in organic acid metabolites [47]. Under salt stress ascorbic acid of Nitraria sibirica was significantly accumulated [57]. Organic acids accumulated under low-temperature stress included 2-isopropylmalic acid, phenylpyruvic acid, 2-propylmalic acid*, 3-isopropylmalic acid*, 2-hydroxyisocaproic acid, 1-aminocyclopropane-1-carboxylic acid, and 2,2-dimethylsuccinic acid. This suggests that malic acid produced by the tricarboxylic acid cycle under low-temperature stress in the *Malus sieversii (Ledeb.) M.Roem.* histoculture seedlings provides raw materials for the synthesis of other metabolites.

The secondary metabolites accumulated by the low-temperature-stressed *Malus sieversii (Ledeb.) M.Roem.* histoculture seedlings mainly included phenolic acids and flavonoids. Phenolic acids include 2,6-dimethoxybenzaldehyde, 5-O-p-coumaroylmangiferyl-O-glucoside, vanillin acetate, raspberry ketone glucoside, 4-O-methyl gallic acid, benzamide, 2-methyl-5-(2-phenylethyl)-1,3-benzenediol, benzyl β-D-glucopyranoside, isochlorogenic acid A*, 1-O-galloyl-2-O-cinnamoyl-β D-glucose, and Fortuneanoside G. Polyphenol biosynthetic pathways are induced especially in plants in response to drought and oxidative stress [54]. Flavonoids include a class of phenylpropanoids that are stored in the vesicles of plant cells as water-soluble pigments [48] and are an ancient and specific class of secondary metabolites in plants, and the accumulation of flavonoids in plant tissues is considered to be a marker of plant adversity [49]. The total flavonoid content increased under salt and alkali stress, but the accumulation of flavonoids was higher under salt stress than alkali stress [58]. Flavonoids accumulated in the leaves of the *Malus sieversii (Ledeb.) M.Roem.* histoculture seedlings under low-temperature stress included epicatechin-4′-O-β-D-glucoside*, gallocatechin 3-O-gallate, kaempferol-3-O-rhamnosyl(12) glucoside, kaempferol-3-O-arabinoside-7-O-rhamnoside, mucoxanthin-7-O-(6″-feruloyl)glucoside, golden sage flavin-6-C-glucoside-4′-O-glucoside, iso-orientin-7-O-(6″-p-coumaroyl)glucoside, dihydroquercetin, naringenin-7-O-glucoside, sacred herbol-7-O-glucoside, 3-hydroxyrhizin, 3-hydroxyrhizin-4′-O-β-D-glucoside, 3-hydroxyrhizin, sacred herbol, ocannin, lignan*, and iso-wild baicalein*. The synthesis of flavonoids increases under environmental stresses such as UV, drought, high salt, and low temperature, which provides an important mechanism to control stress-induced reactive oxygen species (ROS) accumulation [50,51,52,53]. Zanthoxylum bungeanum flavonoids accumulated significantly under drought stress [55]. The amount of flavonoids significantly accumulated under low-temperature stress is the highest compared to other differential secondary metabolism, which suggests that flavonoid metabolites play an important role in the resistance of *Malus sieversii (Ledeb.) M.Roem.* to low-temperature stress.

## 4. Materials and Methods

### 4.1. Succession and Culture of Malus sieversii (Ledeb.) M.Roem. Histoculture Seedlings

*Malus sieversii (Ledeb.) M.Roem.* seeds were obtained from Huocheng County, Yili Kazakh Autonomous Prefecture, Xinjiang. The seeds were laminated and planted in 5 × 10 cavity trays, and when the live seedlings grew to 6–8 pieces, the apical 2–3 cm stem segments were used as explants for tissue culture. The culture medium was prepared by referring to the method of Chen-Chen He [63], and the monoculture clumped shoots with good proliferation were selected for rooting culture every month. After 60 days of culture, the histoculture seedlings with basically the same growth were selected for the experimental treatment, and the histoculture seedlings were cultured in an artificial climate incubator (RXZ intelligent type, Ningbo Jiangnan Instrument Factory and Ningbo City, China) under the following conditions: light intensity of 5000 Lux, daytime 25 °C/14 h, nighttime 23 °C/10 h, and relative humidity of 75%. The −3 °C simulated freezing test treatment was carried out by referring to the method of Fan Zongmin et al. [64] in an artificially modified refrigerator (RongSheng BD/BC-310MS, Hisense Ronshen Refrigerator Co., Ltd. Gongdong, China) with the following conditions: light intensity 5000 Lux, day 25 °C/14 h, night 23 °C/10 h, and relative humidity 75%. The material cultured at 25 °C was used as CK, after which the temperature was cooled down from 25 °C to −3 °C at a rate of 4 °C/h, and the treatment time was 12 h and 36 h (T12h, T36h). The −3 °C treatment was then warmed up to the incubator of 25 °C after T36H to recover for 24 h (HF24h).

### 4.2. Observations on Morphology and Stomatal Structure of Malus sieversii (Ledeb.) M.Roem. Histoculture Seedlings under Freezing Stress

In the experimental treatments, with a total of four treatments, the morphology and stomatal structure of *Malus sieversii (Ledeb.) M.Roem.* were observed and photographed at CK, T12h, T36h, and HF24h. The stomata were classified as open or closed based on the ratio of the length of the line segments (short line length/long line length) marked in the figure (Figure 2a). The stomata were classified as closed stomata when the ratio ≤ 0.3, and open stomata when the ratio > 0.3.

At the time points of CK, T12h, T36h, and HF24h, the stomata of the *Malus sieversii (Ledeb.) M.Roem.* histoculture seedlings leaves were observed. The leaves of the seedlings were cut into 5 mm^2^ squares at each time point, fixed with glutaraldehyde fixative (4%) for 24 h, and then the *Malus sieversii (Ledeb.) M.Roem.* histoculture seedlings were successively dehydrated with varying alcohol concentrations (30%, 50%, 70%, 80%, 90%, and 100%), each gradient being dehydrated for 1 h. Alcohol-soaked (100%) samples were then dried by a supercritical carbon dioxide dryer, and observation of the stomata in leaf blades of the dried *Malus sieversii (Ledeb.) M.Roem.* histoculture seedlings was performed. One field of view was randomly selected at 300× SEM and repeated three times.

### 4.3. Chlorophyll Fluorescence Induction Curve and Chlorophyll Fluorescence Imaging Analysis of Malus sieversii (Ledeb.) M.Roem. Histoculture Seedlings under Freezing Stress

The real-time fluorescence (F) of the leaves of the *Malus sieversii (Ledeb.) M.Roem.* histoculture seedlings was measured using a FMS2 pulse-modulated fluorometer. Real-time fluorescence was stimulated by the measured light, and the intensity of the measured light was gradually increased to reach the intensity that could cause photosynthesis. This is called Actinic Light (AL), Saturation Pulse (SP) technology, which involves turning on a strong light for a very short duration (generally less than 1 s) to temporarily inhibit photosynthesis, and promote chlorophyll fluorescence to reach its maximum value (Fm’). The *Malus sieversii (Ledeb.) M.Roem.* histoculture seedlings were continuously irradiated by Actinic Light, and the light became stronger every 20 s during the process, so that photosynthesis of the *Malus sieversii (Ledeb.) M.Roem.* histoculture seedlings was inhibited.

An IMAGING-PAM modulated fluorescence imaging system (Walz, Effeltrich, Germany) was used. As described previously, the *Malus sieversii (Ledeb.) M.Roem.* histoculture seedlings were placed in the dark at room temperature for 30 min prior to measurements and then placed on a sample stage for chlorophyll fluorescence measurements and to observe Y(NO) imaging. Y(NO) represents unregulated energy dissipation in PS II.

### 4.4. Determination of Metabolites in Leaf of Malus sieversii (Ledeb.) M.Roem. Histoculture Seedlings under Low-Temperature Stress

The data acquisition instrumentation system consisted mainly of ultra-performance liquid chromatography (UPLC) (Shim-pack UFLC SHIMADZUCBM30A) and tandem mass spectrometry (MSMS) (Applied Biosystems 4500 QTRAP). Liquid phase conditions mainly included: (1) column: Waters ACQUITY UPLC HSS T3 C18 1.8 jum, 2.1 mm × 100 mm; (2) mobile phase: ultrapure water (0.04% acetic acid added) for the aqueous phase and acetonitrile (0.04% acetic acid added) for the organic phase; (3) elution gradient: 0 min water/acetonitrile (95:5 V/V), 5:95 V/V for 11.0 min, 5:95 V/V for 12.0 min, and 95:5 for 12.1 min. Based on the MWDB (MetWare database) and metabolite information public database, primary and secondary mass spectrometry analysis was performed according to the existing Mass Bank, KNAPSACK, HMDB, and METLIN mass spectrometry databases, and metabolite quantification was performed using triple quadrupole rod mass spectrometry in multiple reaction monitoring mode. In MRM mode, the quadrupole first screens the parent ion (Q1) of the target substance, excluding the ions corresponding to other molecular weight substances to initially exclude interference; the precursor ions are induced to ionize by the collision chamber and then break to form many fragment ions, which are then filtered and selected by the triple quadrupole. Partial least squares discriminant analysis (PLS-DA) using a multivariate approach maximized metabolomic differences between the two samples, and the relative importance of each metabolite to the PLS-DA model was tested with the variable importance in projection (VIP) parameter. Metabolites with projection parameters ≥1 and difference multiples ≥1.5 or difference multiples ≤0.5 were used as differential metabolites for population discrimination.

### 4.5. Statistical Analysis

The data were analyzed using a one-way analysis of variance (ANOVA) and post hoc multiple comparisons using SPSS v. 19.0 (IBM, Inc., Armonk, NY, USA). Differences were considered to be significant when *p* < 0.05. The values in the figures (SD) (at least *n* = 3).

## 5. Conclusions

The results of study demonstrated that cold stress in *Malus sieversii (Ledeb.) M.Roem.* histoculture seedlings led to wilting, leaf stomatal closure, and photosystem damage. The 1020 metabolites that were found in the leaves of the *Malus sieversii (Ledeb.) M.Roem.* seedlings under freeze stress included amino acids, phenolic acids, nucleotides, flavonoids, lignans, coumarins, sugars, alkaloids, terpenoids, organic acids, lipids, and vitamins. The differential metabolites in the leaves of the *Malus sieversii (Ledeb.) M.Roem.* seedlings grown under low-temperature stress mainly involved glycolysis, amino acid metabolism, lipid metabolism, pyrimidine metabolism, purine metabolism, and secondary metabolites. The *Malus sieversii (Ledeb.) M.Roem.* seedlings responded to the freezing stress by coordinating with each other through these metabolic pathways.

## Figures and Tables

**Figure 1 ijms-25-00310-f001:**
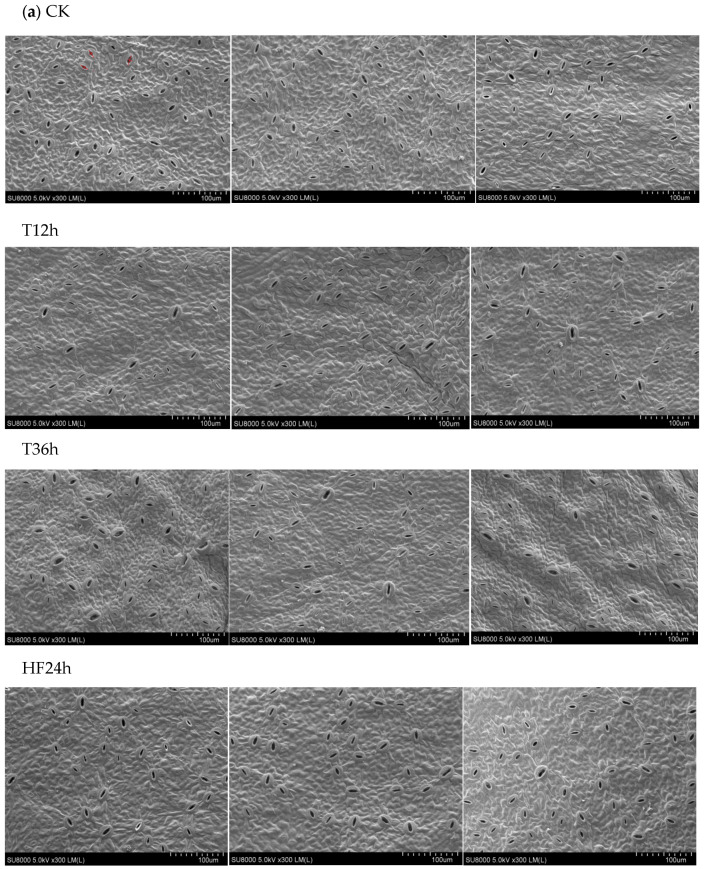
Changes in stomata in *Malus sieversii (Ledeb.) M.Roem.* histoculture seedlings under freezing stress: (**a**) The stomata of *Malus sieversii (Ledeb.) M.Roem.* histoculture seedling leaves in each treatment (CK, T12h, T36h, HF24h). The stomata were classified as closed stomata when the ratio ≤ 0.3, and open stomata when the ratio > 0.3. The red crosses labeled in subfigure a are to determine the opening and closing of the stomata; (**b**) Percentage of the number of open stomata in the leaves of the *Malus sieversii (Ledeb.) M.Roem.* histoculture seedlings in each treatment (CK, T12h, T36h, HF24h). Means denoted by the same letters did not significantly differ at *p* < 0.05 (Duncan’s range test) on a given treatment.

**Figure 2 ijms-25-00310-f002:**
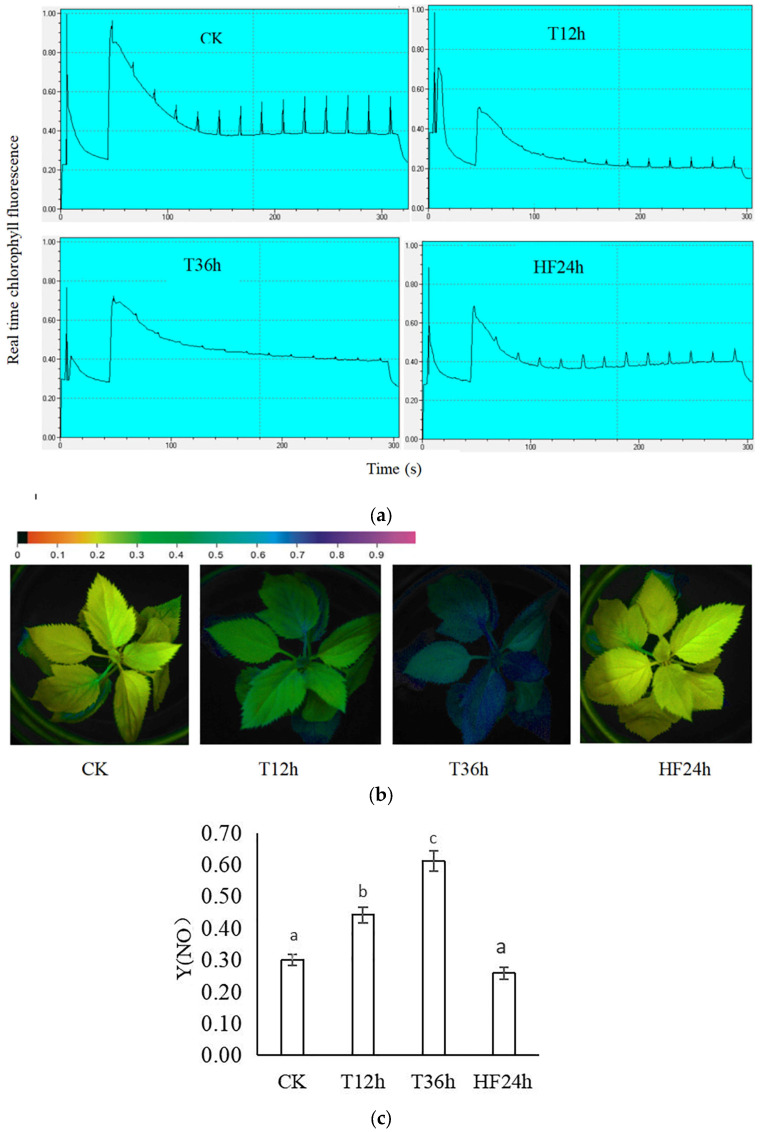
Effect of freezing stress on chlorophyll fluorescence of *Malus sieversii (Ledeb.) M.Roem.* histoculture seedlings: (**a**) Chlorophyll fluorescence induction curves of *Malus sieversii (Ledeb.) M.Roem.* histoculture seedling leaves in each treatment (CK, T12h, T36h, HF24h); (**b**) Y(NO) imaging; (**c**) Value of Y(NO). Means denoted by the same letters did not significantly differ at *p* < 0.05 (Duncan’s range test) on a given treatment.

**Figure 3 ijms-25-00310-f003:**
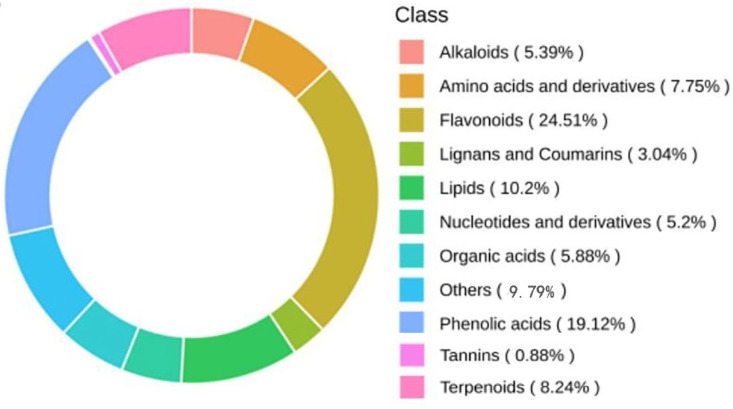
Metabolite class composition ring. Each color represents a metabolite class (where the area of the color block indicates the proportion of that class).

**Figure 4 ijms-25-00310-f004:**
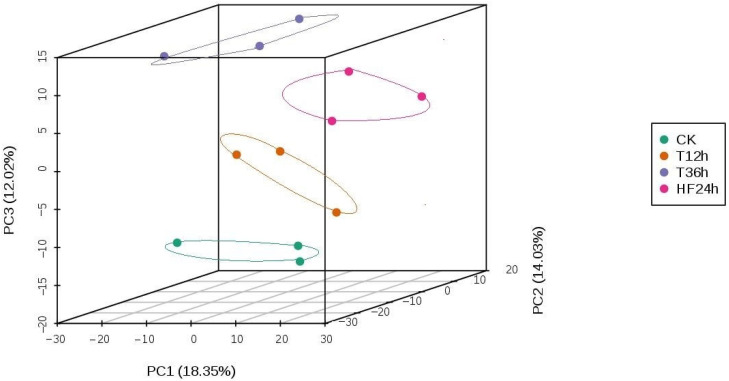
Plot showing PCA scores for each sample data group: PC1 represents the first principal component, PC2 represents the second principal component, PC3 represents the third principal component, and the percentage represents the explanation rate of this principal component to the data set; each point on the graph represents one sample, and samples from the same group are represented by the same color.

**Figure 5 ijms-25-00310-f005:**
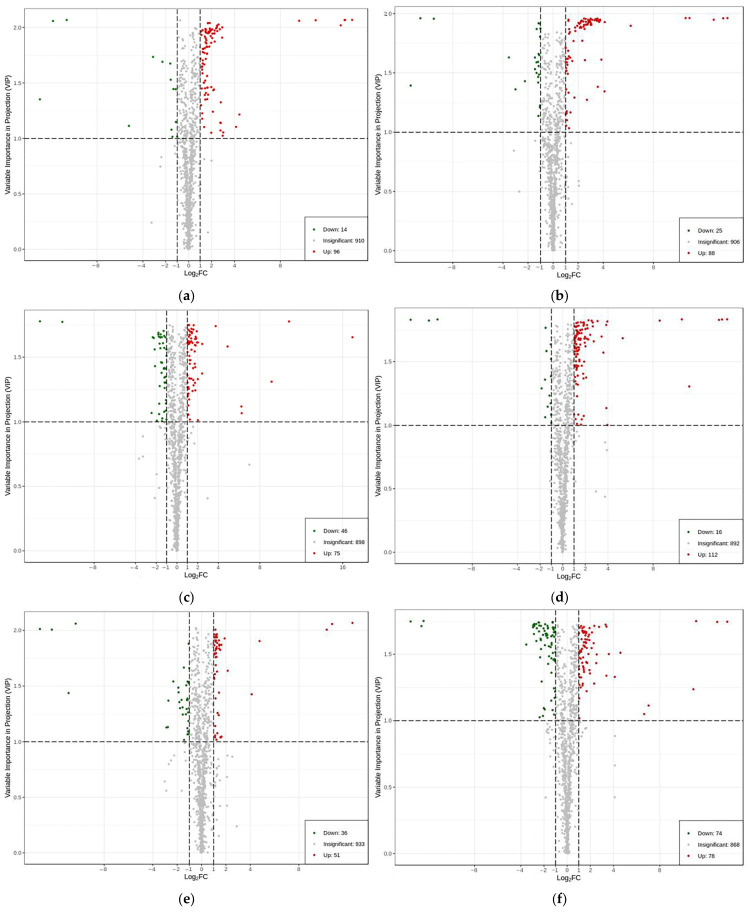
Analysis of differential metabolism and trends in its content: (**a**–**f**) differential metabolites of CKvsT12h, CKvsT36h, T12hvsT36h, CKvsHF24h, T12hvsHF24h, and T36hvsHF24h. The green dots represent downregulated differential metabolites, and the red dots represent upregulated differential metabolites; VIP + FC dual screening conditions: the horizontal coordinate indicates the logarithmic value of the relative content difference of a metabolite in the two groups of samples, and the larger the absolute value of the horizontal coordinate, the greater the relative content difference of the substance between the two groups of samples. The vertical coordinate indicates the VIP value, and the larger the value of the vertical coordinate, the more significant the difference is, and the more reliable the screened differential metabolite is; (**g**,**h**) Venn diagram; (**i**) K-Means clustering diagram, where the horizontal coordinate indicates the sample grouping, the vertical coordinate indicates the standardized metabolite relative content, and the subclass represents the metabolite class number with the same variation trend. The black line is a fit to the trend of relative content of all metabolites in the sub classes.

**Figure 6 ijms-25-00310-f006:**
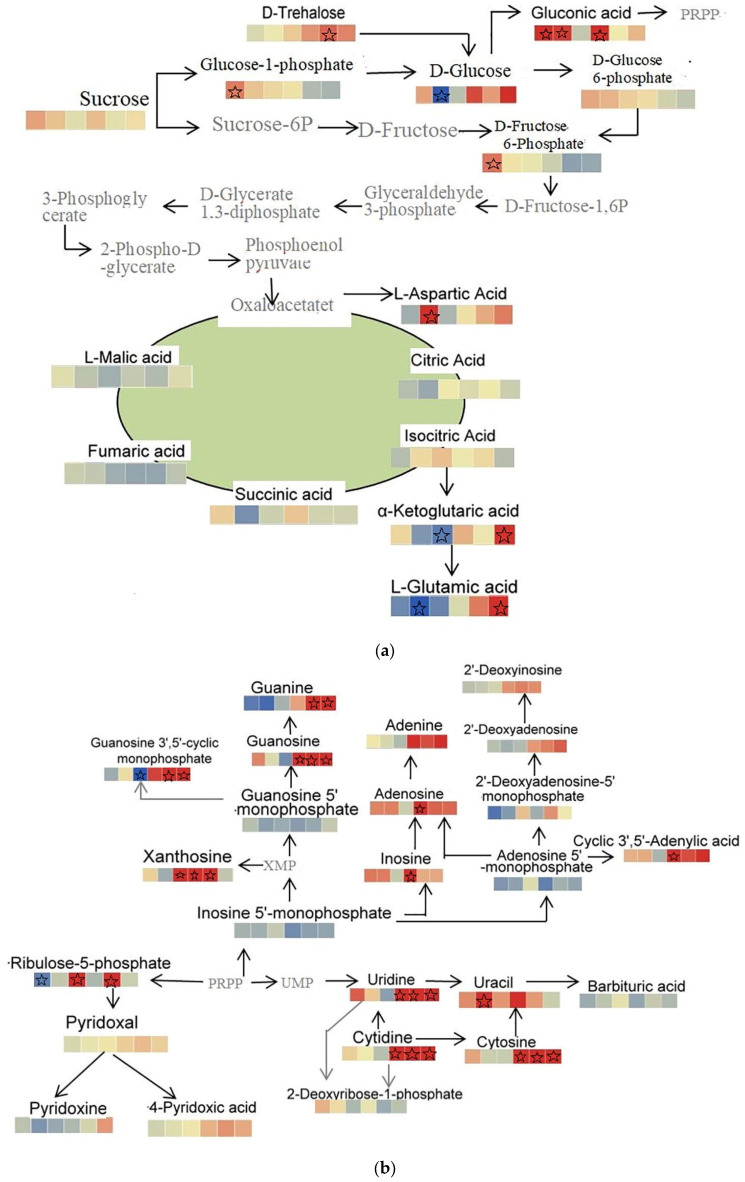
Enrichment analysis of KEGG: (**a**) sugar metabolism; (**b**) purine metabolism, pyrimidine metabolism, and vitamin 6B metabolic pathway; (**c**) amino acid metabolism; (**d**) linolenic and α-linolenic acid metabolism; (**e**) phenolics metabolism, flavonoid metabolism, and xanthone metabolism. The pentagrams are significant differences.

**Table 1 ijms-25-00310-t001:** Number of upregulated and downregulated metabolite classes in each pairwise comparison.

	CKvsT12h		CKvsT36h		CKvsHF24h		T12hvsT36h		T12hvsHF24h		T36hvsHF24h	
	Up	Down	Up	Down	Up	Down	Up	Down	Up	Down	Up	Down
Amino acids and their derivatives	5	0	8	3	12	2	4	2	6	2	8	11
Phenolic acids	5	6	5	4	14	2	7	11	14	3	17	3
Nucleotides and their derivatives	1	0	4	1	19	0	1	2	16	1	16	1
Flavonoids	10	3	13	5	22	6	5	6	6	27	20	14
Lignans and coumarins	5	1	1	2	1	0	1	0	1	1	1	0
Sugars	5	1	2	0	4	0	1	1	1	1	1	1
Alkaloids	3	1	0	3	2	1	0	2	1	1	2	4
Terpenoids	0	3	0	4	3	0	1	2	2	1	1	5
Organic acids	7	0	6	2	10	0	4	1	4	2	4	3
Lipids	61	0	48	0	28	0	9	25	1	27	1	36
Vitamins	0	0	0	1	0	1	0	0	0	1	2	1

## Data Availability

Data is contained within the article and Appendix A.

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
