# Peer review of "Metabolomic Analysis of the Effect of Freezing on Leaves of Malus sieversii (Ledeb.) M.Roem. Histoculture Seedlings"

_ijms, 2023, doi:10.3390/ijms25010310_

Round 1
Reviewer 1 Report
Comments and Suggestions for Authors
The paper may be interesting and has some novel data that merits publication. The problem is that in the present form the manuscript is very poor and contains many mistakes and problems, so needs a major improvement before being suitable for publication.
Text: throughout the text linnean binomial names are not given in italics. The spacing is confusing and sloppy, with spaces before the comma, and no space after the comma, or random double space. That indicates that the manuscript has not been properly edited. Also the format for the citations has not been properly formated. When there is a reference to two papers you should include both numbers in the same bracket. Please correct all these.
Figure legends. Thery are very concise and do not contain the minimum information required to understand the experiment. For instance: figure 1: describe the Y axis in the legend (is %?, so the open stomata are less than 1% in all the samples? in this case the data does not seem to be significative), also describe the X axis and the abbreviations used for each sample.
What does a, and b mean in the bars? Please describe the statistical analysis used and the meaning of the lettering. In figure 2 there is a description of the statistical analysis, but the criteria is different from figure 1, being "a" the lowest value. Please follow the same criteria throughout the whole manuscript.
Figure 3: Parenthesis is missing in the second line of the legend.
Figure 4: Please include a circle indicating the grouping of the samples.
Discussion: Authors refer a lot to what is known for each metabolite, but I have missed a comparison with similar analysis in other woody plants, for instance:
https://bmcplantbiol.biomedcentral.com/articles/10.1186/s12870-018-1464-5
Please, comment.
Comments on the Quality of English Language
needs a revision
Author Response
Dear Reviewer:
Thank you very much for your time and effort in providing valuable comments on my manuscript. I will respond to each of your comments one by one .
Text: throughout the text linnean binomial names are not given in italics. The spacing is confusing and sloppy, with spaces before the comma, and no space after the comma, or random double space. That indicates that the manuscript has not been properly edited. Also the format for the citations has not been properly formated. When there is a reference to two papers you should include both numbers in the same bracket. Please correct all these.
reply;
I have revised the manuscript strictly according to the author's writing instructions .
Figure legends. Thery are very concise and do not contain the minimum information required to understand the experiment. For instance: figure 1: describe the Y axis in the legend (is %?, so the open stomata are less than 1% in all the samples? in this case the data does not seem to be significative), also describe the X axis and the abbreviations used for each sample.
(b)
Figure 1. Changes of stomatal in Malus sieversii(Ledeb.) M.Roem.) histocultures seedlings under freezing stress: (b) Percentage of the number of open stomata in the leaves of Malus sieversii(Ledeb.) M.Roem.) histocultures seedlings in each treatment(CK,T12h,T36h ,HF24h).Means denoted by the same letters() did not significantly differ at P < 0.05 (Duncan’s range test) on a given treatment.
What does a, and b mean in the bars? Please describe the statistical analysis used and the meaning of the lettering. In figure 2 there is a description of the statistical analysis, but the criteria is different from figure 1, being "a" the lowest value. Please follow the same criteria throughout the whole manuscript.
(c)
Figure 2. Effect of freezing stress on chlorophyll fluorescence of Malus sieversii(Ledeb.) M.Roem.) histocultures seedlings:(c) Value of Y(NO) of Malus sieversii(Ledeb.) M.Roem.) histocultures seedlings in each treatment(CK,T12h,T36h ,HF24h).Means denoted by the same letters did not significantly differ at P < 0.05 (Duncan’s range test) on a given treatment.
Figure 3: Parenthesis is missing in the second line of the legend.
Figure 3. Metabolite class composition ring ,each color represents a metabolite class( the area of the color block indicates the proportion of that class).
Figure 4: Please include a circle indicating the grouping of the samples.
Discussion: Authors refer a lot to what is known for each metabolite, but I have missed a comparison with similar analysis in other woody plants, for instance:
During the discussion, similar analytical comparisons of woody plants were made,
- .Hu H, Fei X, He B, Luo Y, Qi Y, Wei A. Integrated Analysis of Metabolome and Transcriptome Data for Uncovering Flavonoid Components of Zanthoxylum bungeanum Maxim. Leaves Under Drought Stress.Front Nutr. 2022 Feb 4;8:801244. doi: 10.3389/fnut.2021.801244.
- 35.Sun S, Fang J, Lin M, Hu C, Qi X, Chen J, Zhong Y, Muhammad A, Li Z, Li Y. Comparative Metabolomic and Transcriptomic Studies Reveal Key Metabolism Pathways Contributing to Freezing Tolerance Under Cold Stress in Kiwifruit. Front Plant Sci.2021 Jun 1;12:628969. doi: 10.3389/fpls.2021.628969.
- 36.Li, H., Tang, X., Yang, X. et al. Comprehensive transcriptome and metabolome profiling reveal metabolic mechanisms of Nitraria sibirica Pall. to salt stress. Sci Rep 11, 12878 (2021). https://doi.org/10.1038/s41598-021-92317-6
- 37.Lu X, Chen G, Ma L, Zhang C, Yan H, Bao J, Nai G, Wang W, Chen B, Ma S, Li S. Integrated transcriptome and metabolome analysis reveals antioxidant machinery in grapevine exposed to salt and alkali stress. Physiol Plant. 2023 May-Jun;175(3):e13950. doi: 10.1111/ppl.13950. PMID: 37291799.
Here, thank you again for your valuable comments on my manuscript, which is of great help to the perfection of my manuscript.

Reviewer 2 Report
Comments and Suggestions for Authors
Dear Authors,
The manuscript „Metabolomic analysis of the effect of freezing on leaves of Malus sieversii histoculture seedlings” was not written according to the guidelines for writing articles. There are a lot of technical errors, and the name of the studied plant is not italicised in most parts of the article. (Ledeb.) M. Roem should be added to the species and the family should be mentioned.
“The results of study demonstrated that cold stress in Malus sieversii histoculture seedling sledto wilting, leaf stomatal closure, and photosystem damage. 1020 metabolites were identifieda…”
1020 metabolites have been identified... this means that a lot of research has been done. Unfortunately, only 4 subsections are listed in Materials and Methods. It is necessary to describe each study on a particular group of metabolites in detail. This would add considerably to the length of the article. I suggest that you present the results of all studies for each group of metabolites in Supplementary materials and write only the summarised results in the Results. This is what research is all about, and the materials and methods consist of four sections.
The article requires considerable additional research and precision in writing. I therefore suggest rejecting the manuscript in this form and recommending that the authors make changes and then submit a new, revised article.
Author Response
Dear Reviewer:
Thank you very much for your time and effort in providing valuable comments on my manuscript. I will respond to each of your comments one by one .
The manuscript „Metabolomic analysis of the effect of freezing on leaves of Malus sieversii histoculture seedlings” was not written according to the guidelines for writing articles. There are a lot of technical errors, and the name of the studied plant is not italicised in most parts of the article. (Ledeb.) M. Roem should be added to the species and the family should be mentioned.
Reply ;
This has been revised in full accordance with the guidelines for writing articles, replacing "Malus sieversisi" with "Malus sieversisi ((Ledeb.) M.Roem.)”
“The results of study demonstrated that cold stress in Malus sieversii histoculture seedling sledto wilting, leaf stomatal closure, and photosystem damage. 1020 metabolites were identifieda…”
1020 metabolites have been identified... this means that a lot of research has been done. Unfortunately, only 4 subsections are listed in Materials and Methods. It is necessary to describe each study on a particular group of metabolites in detail. This would add considerably to the length of the article. I suggest that you present the results of all studies for each group of metabolites in Supplementary materials and write only the summarised results in the Results. This is what research is all about, and the materials and methods consist of four sections.
Reply ;
This study adopts a relatively new research technology-broad target metabolomics, which integrates the advantages of "universality" of non-targeted metabolomics and "accuracy" of targeted metabolomics. It has the characteristics of high throughput, super sensitivity, wide coverage, qualitative and quantitative accuracy, etc. Through MWDB database of Maiwei Company, it can realize one-time qualitative and quantitative detection of 3000+ metabolites in biological samples. Metabolomics can reflect to a greater extent the response of living organisms to external stimuli, pathophysiological changes, stress and provide a new perspective for exploration.
Broad target metabolomics is different from existing metabolite detection methods. Metabolite detection of Malus sieversisi ((Ledeb.) M.Roem.) in vitro seedlings is performed by Maiwei Company, based on MWDB database, and each group of specific metabolites is analyzed. MWDB database is the commercial core data of Maiwei Company, so peak map data related to each specific metabolite cannot be provided. Results Only qualitative analysis and relative quantitative analysis of each metabolite were available.
The article requires considerable additional research and precision in writing. I therefore suggest rejecting the manuscript in this form and recommending that the authors make changes and then submit a new, revised article.
Reply ;
I have revised the manuscript strictly according to the author's writing instructions .
Here, thank you again for your valuable comments on my manuscript, which is of great help to the perfection of my manuscript.

Round 2
Reviewer 1 Report
Comments and Suggestions for Authors
Manuscript has been substantiually improved. I can recommend publication
Author Response
Dear Reviewer:
Thank you very much for your time and effort in providing valuable comments on my manuscript. I will respond to each of your comments one by one .
1.Are all the cited references relevant to theresearch?
Reply;I have removed irrelevant literature and re-added new literature,New literature has been listed
[2]Li,Y.N.Researches of germplasm resources of MalusMill.;China Agriculture Press:Beijing,China,2001;pp.20-134.doi;10.3321/j.issn:0513-353X.2009.03.022.
[4]Xu, J,I.;Liu G.;Ma, P.;Huang Y,P.;Yang, Y,l.;.Research Progress of Adversity Response Mechanism of Malus sieversii.J.Shandong Agricultural Sciences,2014,46(8):4.doi:10.3969/j.issn.1001-4942.2014.08.037.
[5]Chinnusamy V.; Zhu J.; Zhu J.K. Cold stress regulation of gene expression in plants. J.Trends Plant Sci. 2007 .12(10):444-51. doi: 10.1016/j.tplants.2007.07.002.
[8].Zhuang Q.;Chen S.;Jua Z.;Yao Y. Joint transcriptomic and metabolomic analysis reveals the mechanism of low temperature tolerance in Hosta ventricosa.J. PLoS One. 2021. 16(11):e0259455. doi:10.1371/journal.pone.0259455.
[9].Zhang, N.; Zhang, L.; Zhao L.;Ren, Y.;Cui, D.; Chen, J.; Wang, Y.; Yu, P.;Chen, F.iTRAQ and virus-induced gene silencing revealed three proteins involved in cold response in bread wheat.J. Sci. Rep. 2017, 7:7524.doi:10.1038/s41598-017-08069-9.
[10].Clemente-Moreno, N.; Omranian, P.L.; Sáez, C.M.; Figueroa, N.; Del-Saz, M.; Elso, L.; Poblete, I.; Orf, A.; Cuadros-Inostroza, L.A.; Cavieres, L.; Bravo, A.R.; Fernie, M.; Ribas-Carbó, J.; Flexas, Z.; Nikoloski, Y.; Brotman, J.Low-temperature tolerance of the Antarctic species Deschampsia antarctica: a complex metabolic response associated with nutrient remobilization.J.Plant Cell Environ. 2020.43.1376-1393.doi:10.1111/pce.13737.
2.Are the results clearly presented?
结果是否清晰呈现?
2.2. Effect of freezing on chlorophyll fluorescence of Malus sieversii(Ledeb.) M.Roem.histocultures seedlings
Chlorophyll fluorescence induction curves were observed in Malus sieversii (Ledeb.) M.Roem. histocultures seedlings. When the real-time fluorescence of Malus sieversii (Ledeb.) M.Roem. histocultures seedlings stabilized, the jump in maximum fluorescence (Fm') excited by strong light (saturating pulse) weakened with the prolongation of the freezing stress time (CK,T12h, T36h); there was a significant recovery at HF24h (Fig. 2a),
(Changes in the manuscript have been highlighted in red.)
2.3.2 Qualitative Analysis of leaf metabolites in Malus sieversii(Ledeb.) M.Roem. histocultures seedlings under freezing stress
According to the OPLS-DA model, the metabolite group data were analyzed to plot the scores of each group (Figure S1) and further show the differences between the groups.The prediction parameters of the OPLS-DA evaluation model were R2X, R2Y and Q2, R2X and R2Y represent the explanation rate of the model for the X and Y matrices, respectively, and Q2 indicates the predictive ability of the model. The closer these three indicators are to 1, the more stable and reliable the model is. Q2>0.5 can be regarded as an effective model, and Q2>0.9 is an excellent model, In each OPLS-DA evaluation model, Q2 and R2Y were greater than 0.9, and R2X was greater than 0.5,which shows that the model is reliable. From the results, there are obvious differences between the four groups of data(CK,T12h,T36h and HF24h) (Figure S2). The differentiated metabolites could be screened according to the VIP value analysis (Figure S3).
(Changes in the manuscript have been highlighted in red.)
Here, thank you again for your valuable comments on my manuscript, which is of great help to the perfection of my manuscript.
Reviewer 2 Report
Comments and Suggestions for Authors
The species name should be written this way: Malus sieversii (Ledeb.) M.Roem.
The name of the genus and species should be written in italics.
The article requires a detailed review before publication.
Author Response
Dear Reviewer:
Thank you very much for your time and effort in providing valuable comments on my manuscript. I will respond to each of your comments one by one
Comments and The species name should be witten this way: Malus sieversi (Ledeb.) M.Roem.
Suggestions for Authors
Reply;Replace Malus sieversi (Ledeb.) M.Roem.) with Malus sieversi (Ledeb.) M.Roem.
The name of the genus and species should be witten in italics.
The article requires a detailed review before publication.
Reply;I've had a detailed examination.
Here, thank you again for your valuable comments on my manuscript, which is of great help to the perfection of my manuscript.